# Gender Differences in Sexual Information Needs and Relating Factors in Cancer Patients: A Cross-Sectional Survey

**DOI:** 10.3390/ijerph18073752

**Published:** 2021-04-03

**Authors:** Hae Won Kim, Yeon Hee Kim, Saem Yi Kang, Eun Ju Lee, Jung Lim Lee, Youngji Kim

**Affiliations:** 1Center for Human-Caring Nurse Leaders for the Future by Brain Korea 21 (BK 21) Four Project, Department of Nursing, The Research Institute of Nursing Science, College of Nursing, Seoul National University, Seoul 03080, Korea; haewon@snu.ac.kr; 2Department of Clinical Nursing, University of Ulsan, Ulsan 44610, Korea; kimyhee@amc.seoul.kr; 3College of Nursing, Seoul National University, Seoul 03080, Korea; rkdtoadl@snu.ac.kr; 4Department of Nursing, Inha University, Incheon 22212, Korea; ariggari@snu.ac.kr; 5Department of Nursing, Daejeon University, Daejeon 34519, Korea; jlim0202@hanmail.net; 6Department of Nursing, College of Nursing and Health, Kongju National University, Gongju-si 32588, Korea

**Keywords:** sexuality, health, nurses, practice, oncology

## Abstract

This study aimed to identify the sexual information requirements and related factors according to gender to improve sexual health in cancer patients. In this cross-sectional study, a total of 687 cancer patients from a single cancer center in Korea completed a self-reported questionnaire. Multiple logistic regression analysis was used to compare the sexual information requirements and related factors among cancer patients. The results showed that male cancer patients had higher demands for sexual information than female cancer patients (t = 27.11, *p* < 0.001). Men appeared to have a greater need than women in the need for sexual information (t = 30.41, *p* < 0.001) and professional sexual intervention (t = 21.97, *p* < 0.001). Regarding sexual information needs, income (OR: 0.43, 95% CI: 0.25 to 0.73) was a significant factor in men. In women, age (OR: 0.51, 95% CI: 0.31 to 0.86), alcohol consumption (OR: 1.88, 95% CI: 1.12–3.16), and chemotherapy (OR: 1.87, 95% CI: 1.12–3.12) were significant factors. Significant differences in the overall sexual information needs and related factors were observed between male and female cancer patients. Therefore, a strategy needs to be established to improve the sexual health of cancer patients considering gender differences.

## 1. Introduction

Cancer is among the leading causes of death worldwide. Worldwide, an estimated 19.3 million new cancer cases and almost 10.0 million cancer deaths occurred in 2020 [1]. In the Republic of Korea, the number of new cases of cancer was 232,255 and the five-year life rate after cancer was less than 50 percent in 1990, though a recent survey showed that the rate was nearly 70 percent [2]. With the increase in cancer survival rate, healthcare professionals’ interest is drawing on life after cancer treatment.

Recent research has shown that cancer may affect the sexual function and sexual activity of cancer patients, thus affecting their quality of life [3,4,5]. Sexual health does not mean that there is no disease or impairment, but rather a physical, emotional, mental, and social wellbeing in relation to sexuality [6]. This is a major concern for cancer patients and is an important determinant of the treatment compliance of cancer patients [7,8]. Hence, practical interventions are needed to improve the sexual health of cancer patients.

Previous studies have found that the most common sexual problems for people after cancer treatment are a decrease in sexual frequency, sexual satisfaction, and engagement for sexual activity in both men and women; problems achieving and maintaining an erection in men; and pain with intercourse in women [9]. Previous studies have reported that approximately 35% of all cancer survivors wanted more information about sexual health and cancer survivors’ sexual concern were different between male and female cancer patients. Male cancer survivors were most concerned about being able to satisfy their partners, while female cancer survivors were most concerned with changes in their body image in sexuality [10]. These studies showed that males and females have commonalities and differences in sexual behavior and perceptions of cancer survivors. However, most of the research has mainly focused on women who have breast or gynecologic cancer and men who have prostate cancer [11,12]. Few studies have considered the gender issue in the sexual health of all types of cancer patients.

Sexuality problems of cancer patients can be addressed through effective communication with healthcare professionals (HCPs). A HCPs’ understanding of sexual problems in cancer patients is often too limited to meet the patients’ expectations [13]. Most cancer patients are interested in the changes in sexual function and sexual activity that may occur after treatment. They want support and methods to overcome sexual changes [14]. On the other hand, cancer patients feel that there is insufficient explanation of the sexual problems and sex-related intervention [15]. Moreover, cancer patients are reluctant to communicate their sexuality to healthcare providers [16], suggesting sexual counseling and effective communication with healthcare providers should be provided. To develop sexual health interventions to meet the patients’ needs, the sexual information required by patients should be identified, and effective communication should be developed [17,18].

Factors affecting the sexual health of cancer patients include demographics, such as age and gender, psychological factors, cancer type, and chemotherapy [19,20]. In particular, the sexual interests, sex-related symptoms, and sex-seeking behaviors may differ between males and females owing to the differences in male and female reproductive organs and sociocultural differences [21,22,23,24]. Therefore, to improve the sexual health of cancer patients, differentiated counseling and education will be possible for each gender if the health information needs according to gender differences can be identified.

Research has been actively conducted on sexual health nursing in cancer patients. Despite this, the reported studies were conducted to identify the sex-related problems of patients with specific cancer types and the attitudes and views of HCPs [25]. On the other hand, sexual health nursing is still not included in the general care of cancer patients, and the information needs according to gender differences have not been identified. This study examined the sexual information needs according to gender to improve the sexual health of cancer patients. In addition, this study examined the gender-related communication barriers in sex-related communication experienced by cancer patients. The results of this study can be used as a basis for the development of strategies for promoting cancer patient-centered health.

## 2. Materials and Methods

### 2.1. Study Design

This single-center study was undertaken in Korea as a cross-sectional descriptive survey.

### 2.2. Study Participants

The participants were diagnosed with cancer and had visited a cancer center for a follow up in one of the tertiary hospitals in Seoul. The cancer centers operate outpatient services for cancer survivors after anticancer treatment. The study sample consisted of patients aged between 20 years and 75 years, who had completed anticancer therapy, could read Korean, and could provide informed consent. Exclusion criteria included having a major psychiatric illness, with limited life expectancy due to debilitating medical conditions, and metastatic disease. Final screening and recruitment were conducted by researchers. Considering previous studies [26], under the condition of odds ratio 1.3, two-sided testing, significance level of 0.05, and power of 0.80, the sample size calculated using G power 3.1 was 620 people, and 1000 participants were recruited considering the dropout rate of 35%. They were selected through convenience sampling.

### 2.3. Recruitment and Data Collection

The recruitment period was from February to May 2015. A total of 1000 questionnaires were distributed. Seven hundred subjects who understood the purpose of the study and agreed to participate were enrolled in this study. Of these, 690 respondents completed the written informed consent form, resulting in a 98.6% response rate. After excluding three subjects who did not answer or fill in a blank, 687 subjects were finally included in the analysis. One of the researchers (YHK) trained research staff to seek participants’ consent and collect data for this study. Research staff with more than three years of clinical nursing experience explained the purpose and procedure of this study to the subjects who then agreed to participate. The questionnaire was then distributed to the subjects in a separate room at the outpatient department. The time required to complete the questionnaire was approximately 10 min. There was no incentive to participate in this study.

### 2.4. Instrument

The questionnaire consisted of the following: (1) demographics and disease characteristics of the subjects, (2) sexual information needs, and (3) feelings during sexual communication and characteristics related to sexual information.

#### 2.4.1. Demographics and Disease Characteristics

A brief demographics questionnaire was given to the participants, which consisted of questions that asked them to report their age, sex, sexual orientation, relationship status and duration, education level, and their perception of physical health.

#### 2.4.2. Sexual Information Needs

Sexual information needs refer to the degree of information seeking related to sexual life that a cancer patient requires. Sexual information needs were measured with nine items of a tool developed in a prior study [27]. This tool consisted of three factors: five items for the necessity of sexual information (need 1), two items for sexual satisfaction (need 2), and two items for professional sexual intervention (need 3). Regarding the reliability of the published tool, the Cronbach’s α value at the time of tool development was 0.83 [19]. In this study, the Cronbach’s α value of this tool was 0.86. Regarding the reliability of the three factors of this tool, the Cronbach’s α values were 0.87, 0.70, and 0.86, respectively. All items were assessed with a five-point Likert scale ranging from one (not at all) to five (very much). The total score ranged from 9 to 45, with a higher score indicating a greater need for information on sexual life.

#### 2.4.3. Feelings during Sexual Communication and Characteristics Related to Sexual Information

Three items were examined to investigate the emotions related to sexual communication: shame, embarrassment, and discomfort during a conversation with medical professionals. All items were assessed with a five-point Likert scale ranging from one (not at all) to five (very much), with a higher score indicating greater perceived negative feelings during communication with health professionals. The following items were surveyed to investigate the experience of sex education: sex education-related experiences, sexual education with their partner, and sex information sources.

### 2.5. Ethical Considerations

Ethical approval for this study was granted by the Institutional Review Board (IRB) of S University (IRB approval No. S2013-1995-0005). The authors respected the voluntary intention of the research participants to take part in this research. Written consent was obtained only when the participant understood the contents of this study and agreed to participate. It was explained to participants that they could withdraw their intention to participate at any time. They were also informed that would be no disadvantage if they did not participate in this study or dropped out. The questionnaire responses were anonymized; they were not used for other purposes.

### 2.6. Data Analysis

All statistical analyses were performed using the Statistical Package for Social Sciences (version 23, SPSS Statistics, IBM Corp, Chicago IL, USA, 2016). The statistical significance was set to *p* < 0.05. The general characteristics and disease characteristics of the cancer patients were analyzed using descriptive statistics and percentages. The sex education-related experiences and preferred types according to gender were analyzed by Chi-square analysis. The sexual information needs and feelings during sex-related communication were compared using a t-test. For both groups, the effects of the sociodemographic, clinical factors, and feelings during sex-related communication were analyzed using an analysis of Spearman’s rho. We used multivariate logistic regression to examine sex informational needs after cancer treatment. This estimates the probability of mutually exclusive events and, hence, is most often used with dichotomous dependent variables, as in this case with sex informational needs (0 = low, 1 = high). To evaluate the goodness of fit of the logistic regression model, we calculated Nagelkerke’s R square. The Nagelkerke’s R squared indicates the power of explanation of the model. The significant factors from Spearman correlation analysis were subjected to logistic regression analysis using a backward conditional procedure for multivariate analysis to establish the logistic regression model. For men, the monthly income, embarrassment, and displeasure variables were used, and for women, age, alcohol intake, chemotherapy, embarrassment, and displeasure variables were used in the logistic regression model. The odds ratios (ORs) and confidence intervals (CIs) were calculated to examine the impact of variables on the sexual information needs.

## 3. Results

### 3.1. General Characteristics of the Study Participants

A total of 687 patients participated in the study (male, *n* = 350, 50.95%; female, *n* = 337, 49.05%). The participants were aged between 26 and 74 years, with a mean age of 52.84 ± 10.57 years for males and 48.70 ± 9.41 years for females. Based on the distribution of the subjects’ diagnoses, gastric cancer, colon cancer, and liver cancer were the most common cancers in men. In contrast, breast cancer, colon cancer, and gastric cancer were the most common cancers in women. The subjects’ cancer distribution results were similar to the national prevalence of cancer [28] (Table 1).

Regarding previous sex education, there were no significant differences in attending sex education and the preference for sex education with a sexual partner between men and women. Regarding the sources used to obtain sexual information by the participants, the Internet (22.3%) had the highest percentage as a source, followed by television (14.6%) and books (10.4%) for men. For women, television (11.7%) had the highest percentage, followed by the Internet (10.2%) and patient education (9.3%). When asked who would be the preferred person to provide sex education if they had the opportunity, a doctor had a higher rank (33.1% in men and 20.8% in women) than a nurse (4.7% in men and 15.2% in women). Regarding the preferred place for sex education, men favored a hospital counseling room (56.9%) and online education (13.3%), and women favored a hospital counseling room (65.4%) and patient peer meeting (10.1%). When asked about what type of sex education that they would like to receive, there was a significant difference in the answers (χ^2^ = 15. 24, *p* < 0.001) between men and women. For men, the most preferred type was the private type (51.9%), but the group type was favored by women (35.5%) (Table 2).

### 3.2. Sexual Information Needs of Patients with Cancer and Feelings during Sex-Related Communication with HCP

Significant differences were observed in the global scores of sexual information needs (t = 27.11, *p* < 0.001), suggesting that men desired more sexual information than women. Of the three subscales, men appeared to have greater needs for the “necessity of sexual information” (t = 30.41, *p* < 0.001) and “professional sexual information” (t = 21.97, *p* < 0.001) than women. Although there were differences in sexual satisfaction between men and women, these differences were statistically insignificant. When asked, “What do you think of when talking about sexual issues with a healthcare practitioner?” male cancer patients tended to respond that a HCP would be unpleasant compared to women (Table 3).

### 3.3. Relationship between Sexual Information Needs and General Characteristics

Regarding correlations between the sexual information needs and general characteristics, the sexual information needs of men were correlated significantly with monthly income (r = −0.17, *p* = 0.008), “I would be embarrassed” (r = 0.12, *p* = 0.042), and “Health professional would be unpleasant” (r = 0.13, *p* = 0.027). In contrast, the sexual information needs of women were significantly correlated with age (r = −0. 23, *p* < 0.001), alcohol drinking (r = 0.20, *p* = 0.001), chemotherapy (r = 0.17, *p* = 0.005), “I would be embarrassed” (r = 0.13, *p* = 0.033), and “Health professional would be unpleasant” (r = 0.13, *p* = 0.032) (Table 4).

### 3.4. Factors Affecting the Sexual Information Need of Patients with Cancers Identified by Logistic Regression Analysis

Regarding the sexual information, the monthly income (OR: 0.43, 95% CI: 0.25 to 0.73) was a significant factor in men. In women, age (OR: 0.51, 95% CI: 0.31 to 0.86), alcohol consumption (OR: 1.88, 95% CI: 1.12–3.16), and chemotherapy (OR: 1.87, 95% CI: 1.12–3.12) were significant factors (Table 5).

## 4. Discussion

Sexual health is an essential part of the quality of life in cancer patients. This study evaluated the sexual information needs and affecting factors between male and female cancer patients to obtain basic data for the development of strategies for sexual health promotion.

In this study, cancer patients had moderate levels of sexual information needs and perceived sexual information as an important issue in cancer care. This result was similar to previous results in that sexual wellbeing was rated as important by more than two-thirds of cancer patients [29]. Although one-third of participants wanted to receive sexual information, approximately 5% of the participants had been educated about sexual life during the cancer disease trajectory, suggesting that the information needs of cancer patients were not being met. Previous studies have reported that cancer patients have unmet needs regarding information about their sex life after cancer therapy, and only 37% of cancer patients received sufficient sexual information regarding cancer treatment [30,31]. This suggests that sexual healthcare must be included in the current cancer regimen.

In addition, this study found that the sexual information needs of male cancer patients were higher than those of female cancer patients, and men had greater needs for medical intervention than women. This result is consistent with the results of previous studies showing that female cancer patients had lower rates of consultation with hospital staff regarding sexual life than male cancer patients [32,33]. Kim and Kim [32] reported that 21% of male cancer patients communicated with HCPs regarding sex-related changes, whereas only 4.8% of female cancer patients did. On the other hand, Gilbert et al. [33] found that approximately 95% of subjects had a consultation with a healthcare provider about sex-related problems in a Caucasian Australian study (68% in men and 43% in women). In the present study, female cancer patients were less likely to have sex-related information needs, and they might have little chance to communicate with HCPs if the HCP did not enquire about sexual issues with their clients. They were identified to be passive in seeking sex-related information compared to men. This might be due to the indifference of medical personnel [34], lack of sex education, lack of sexual autonomy, cultural factors, and embarrassment with communication [21,35]. Therefore, HCPs need to support female cancer patients to recognize their sexual needs and explore the reasons why female cancer patients are passive in expressing their needs and neglect asking for professional help. This requires in-depth research into how female cancer patients perceive their sexual needs in a variety of sociocultural contexts.

This study showed that factors affecting sexual information needs differed according to gender. In addition, male cancer patients had lower sexual information needs when they had a higher income. In contrast to the present study, previous studies showed a decrease in sexual interest among those with low income and low education [36,37]. Further studies will be needed to clarify this. In this study, the average age of male subjects was 52 years old, and their sex information needs tended to decrease with increasing income. A high-income job may lead to high stress and high fatigue, thereby reducing the interest in sex [38].

Unlike men, age was the most important variable for female cancer patients. The need for sexual information in women decreased with age and increased with more alcohol consumed. In a previous study, a lower age was associated with a higher demand for sex-related information [39]. In this study, in the case of females, the older the age, the more likely that sexual needs would be suppressed. In addition, in the case of female cancer patients who consumed alcohol, the likelihood of expressing their sexual desires or wishes relatively easily is known to be increased, with the increasing amount of alcohol they drank [40]. Therefore, we assumed that the demand for sex-related information was greater. Female cancer patients showed an increased need for sexual information after they received chemotherapy. In previous studies, after receiving chemotherapy, female cancer patients had experiences of impaired sexuality, decreased sexual interest, marital conflict, and a sense of self-defeating concern about their sexual life [41,42]. These findings support the need for nursing intervention in female cancer patients, with a consideration of related factors (age, chemotherapy, and alcohol consumption). In other words, among the female cancer patients, sex education should be provided according to the characteristics of patients, as the demand for sex information is high in patients with younger age, chemotherapy, or increased alcohol intake. Due to the fact that the interest in sex information from cancer patients increases according to the characteristics of the patient, it is necessary to support the emotional stability of the patient by providing this information during the cancer patient’s family counseling.

The results showed the type and modality of sexual information access that the cancer patients preferred. Most of the cancer patients in this study desired sex education in hospital. While male patients preferred individual education, female patients preferred group education. Therefore, is necessary to make various combinations of education times and cycles, education methods, and education places that patients can easily access and talk comfortably about their troubles and obtain counseling. Most of the participants required professional sexual intervention. On the other hand, they expected that health professionals would be uncomfortable when having sexual conversations with them. The readiness of HCPs in sexual care is also an essential element for sex education that is suitable for cancer patients. Wei et al. [43] reported that cancer patients require hope for survival and more positive information for the future. In contrast, the HCPs thought patients wanted more information on efficacy and safety, suggesting a discrepancy in the perspectives between patients and medical staff. Reese et al. [44] also reported that sex-related communication is provided less actively and less frequently in regards to the patients’ sexual needs. In addition, the patients experience barriers when medical personnel consult with them regarding their sexual life [21]. HCPs complain of a lack of knowledge regarding sexual life and a sense of inconvenience during consultation [43,45]. Thus, HCPs require sex-specialized education. In addition, there is a need for in-depth research into factors that may interfere with the delivery of sex-related health services.

The findings of the present study revealed that a substantial number of cancer survivors perceived information about sexuality to be an important issue, but they had not received enough information regarding sexuality. In addition, it showed that there was a communication barrier between cancer survivors and the HCPs regarding sexuality. This study showed that there were some differences between the genders, specifically, lesser sexual information needs among the females. The reason for the gender difference is unknown, but it could be due to talking about sexuality being considered as “uncomfortable” by patients [16], and inadequate sex knowledge [46]. It is important to explore how knowledgeable patients are about sexuality in cancer rehabilitation for the nurse to be able to provide adequate information, intervention, and support to patients and their partners.

This study helps HCPs recognize that cancer patients require sexual information and professional intervention. This study highlighted the ongoing need within the nursing care of sexual dysfunction in women after cancer treatment. HCPs should include sexual health as a routine component of assessments to all providers caring for cancer patients so that patients can more comfortably share their concerns. Careful intervention is needed to identify the female cancer patients’ sexual needs, and HCPs should be sensitive to the sexual needs of cancer patients who require professional intervention. Also, after cancer treatment, cancer survivorship programs including sexual health care should be launched in the public health care system for follow up [47].

In this study, convenience sampling revealed differences in the demographic characteristics of men and women, such as age, income, smoking, and drinking, which should be considered when attempting to generalize the findings. Although the correlation between the variables was significant, the relationships identified between variables were extremely weak. It presumed that the dependent variables were not normally distributed. The limitation of this study is that data were collected in 2015, not current. However, it can be seen as an advantage that we dealt with the sexuality of cancer patients with all types of cancer.

## 5. Conclusions

Cancer patients require information on sexuality and professional intervention for their sexual health. Significant differences in the overall sexual information needs and related factors were observed between male and female cancer patients. It is important to assess how knowledgeable patients are about sexuality in cancer rehabilitation in order for HCPs to be able to provide adequate information, intervention, and support to cancer survivors. Therefore, a strategy needs to be established to improve the sexual health of cancer patients considering gender differences.

## Figures and Tables

**Table 1 ijerph-18-03752-t001:** Demographic characteristics and disease characteristics of participants (*n* = 687).

Variable	Categories	Men (*n* = 350)	Women (*n* = 337)	χ^2^ (*p*)
		*n* (%)	*n* (%)	
Age (*n* = 682)	20–40	48 (13.8)	69 (20.7)	37.19 (<0.001)
	41–50	72 (20.6)	114 (34.2)	
	51–60	147 (42.1)	117 (35.1)	
	61–74	82 (23.5)	33 (9.9)	
Job status (*n* = 651)	No	149 (44.1)	224 (71.6)	57.59 (<0.001)
	Yes	189 (55.9)	89 (28.4)	
Monthly income (Korean 10,000 won, *n* = 492)	3–200	54 (20.7)	38 (16.5)	2.85 (0.414)
	201–400	105 (40.2)	92 (39.8)	
	401–600	67 (25.7)	73 (31.6)	
	601–3000	35 (13.4)	28 (12.1)	
Marital status (*n* = 675)	Married	296 (85.5)	274 (83.3)	0.66 (0.240)
	Unmarried, etc.	50 (14.5)	55 (16.7)	
Smoking (*n* = 673)	No	49 (14.1)	303 (92.9)	418.63 (<0.001)
	Yes	298 (85.9)	23 (7.1)	
Alcohol drinking (*n* = 648)	No	70 (21.1)	165 (52.2)	67.88 (<0.001)
	Yes	262(78.9)	151(47.8)	
History of surgery (*n* = 656)	No	222 (65.3)	152 (48.1)	19.75 (<0.001)
	Yes	118 (34.7)	164 (51.9)	
History of genital surgery (*n* = 687)	No	334 (95.4)	295 (87.5)	13.83 (<0.001)
	Yes	16 (4.6)	42 (12.5)	
History of chemotherapy (*n* = 668)	No	152 (44.2)	142 (43.8)	0.009 (0.938)
	Yes	192 (55.8)	182 (56.2)	
History of radiation therapy (*n* = 664)	No	275 (80.6)	230 (71.2)	8.11 (0.005)
	Yes	66 (19.4)	93 (28.8)	
Diagnosis (*n* = 664)	Gastric cancer	71 (20.3)	30 (8.9)	
	Colon cancer	59 (16.9)	44 (13.1)	
	Liver cancer	36 (10.3)	6 (1.8)	
	Breast cancer	0 (0)	134 (39.8)	
	Lung cancer	29 (8.3)	9 (2.7)	
	Reproductive organ cancer	9 (2.6)	19 (5.6)	
	Others	146 (41.7)	95 (28.2)	

**Table 2 ijerph-18-03752-t002:** Preferred type and modality of sex education.

Characteristics	Categories	Men (*n* = 350)	Women (*n* = 337)	χ^2^ (*p*)
		*n* (%) or mean ± SD		
Experience of attending to sex education (*n* = 643)	No	311 (95.1)	297 (94.0)	0.39 (0.603)
	Yes	16 (4.9)	19 (6.0)	
I want to receive sex education with sexual partner (*n* = 648)	No	228 (68.5)	224 (71.1)	0.54 (0.494)
	Yes	105 (31.5)	91 (28.9)	
Where did you get the information related to sex? (*n* = 332, multiple response; 528)	Internet	118 (22.3)	54 (10.2)	
	TV	77 (14.6)	62 (11.7)	
	Book	55 (10.4)	22 (4.2)	
	Patient education (hospital)	48 (9.1)	49 (9.3)	
	Patient peer meeting	15 (2.8)	25 (4.7)	
	Friend	2 (0.4)	1 (0.2)	
(1) If you think sex education would be necessary, who would be the favorable person for education? (*n* = 324, multiple response; 341)	Doctor	113 (33.1)	71 (20.8)	
	Nurse	16 (4.7)	52 (15.2)	
	Social worker	14 (4.1)	7 (2.1)	
	etc.	30 (8.8)	38 (11.1)	
(2) Which place or ways do you prefer for sex education? (*n* = 343, multiple response; 360)	Hospital counseling room	103 (56.9)	117 (65.4)	
	Online	24 (13.3)	12 (6.7)	
	Outpatient department	14 (7.7)	8 (4.5)	
	Outside hospital	12 (6.6)	2 (1.1)	
	Patient peer meeting place	8 (4.4)	18 (10.1)	
(3) Which type of sex education do you want? (*n* = 411)	Private	108 (51.9)	67 (33.0)	15.24 (<0.001)
	Group	50 (24.0)	72 (35.5)	
	Don’t know	50 (24.0)	64 (31.5)	

**Table 3 ijerph-18-03752-t003:** Comparison of the sexual information needs and feeling during sex-related communication with HCP.

Items	Dimensions	Men (*n* = 350)	Women (*n* = 337)	*t*	*p*
		Mean ± SD			
Sexual information needs	**Global scores**	27.89 ± 5.60	25.59 ± 5.23	27.11	<0.001
	**Need for sexual information**	16.39 ± 4.12	14.60 ± 4.07	30.41	<0.001
	1. Information about sex or sexuality is important	3.74 ± 0.88	3.45 ± 0.96	15.88	<0.001
	2. I very much want to get the sexual information	3.29 ± 1.08	3.03 ± 1.05	9.57	0.002
	3. I have curiosity about sex or sexuality	3.14 ± 1.07	2.75 ± 0.98	23.64	<0.001
	4. I want counselling about my sexuality from a professional	2.71 ± 1.07	2.48 ± 0.95	8.57	0.004
	5. Sex or sexuality is important in my life	3.52 ± 0.99	2.87 ± 1.02	67.17	<0.001
	**Sexual satisfaction**	6.45 ± 1.72	6.26 ± 1.76	1.95	0.163
	6. My sexual partner is satisfied with the current sex	3.18 ± 0.95	3.09 ± 0.93	1.32	0.250
	7. My sexual partner is honest in talking about sex problems	3.27 ± 1.04	3.16 ± 1.05	1.72	0.191
	**Professional sexual intervention**	5.94 ± 1.95	5.22 ± 1.96	21.97	<0.001
	8. I want a health professional to be able to initiate to ask about sexual issues	2.98 ± 1.02	2.70 ± 1.03	12.45	<0.001
	9. I want a health professional to be able to solve sexual problem	2.95 ± 1.03	2.52± 1.02	28.76	<0.001
Feeling during sex-related communication with HCP	Shame (I would be shameful)	2.32 ± 0.94	2.40 ± 0.95	1.12	0.290
	Embarrassment (I would be embarrassed)	2.37 ± 0.97	2.44 ± 0.96	1.00	0.319
	Unpleasure (Health professional would be unpleasant)	2.52 ± 0.92	2.35 ± 0.89	6.27	0.013

**Table 4 ijerph-18-03752-t004:** Correlations among sexual information needs, general characteristics and feeling during sex-related communication with a healthcare professional (HCP).

Variables	Total Sexual Information Needs Spearman’s Rho (*p*)	
	Men	Women
Sociodemographic and clinical factors		
Age	0.09 (0.874)	−0.23 (<0.001)
Religion	0.002 (0.972)	−0.01 (0.899)
Job status	−0.04 (0.494)	0.10 (0.085)
Monthly income	−0.17 (0.008)	0.01 (0.911)
Marital status	−0.03 (0.552)	−0.02 (0.748)
Smoking	0.10 (0.071)	0.06 (0.322)
Drinking	0.04 (0.539)	0.20 (0.001)
Time since diagnosis	0.01 (0.826)	0.05 (0.415)
Surgery	0.06 (0.295)	−0.03 (0.581)
Genital operation	0.31 (0.260)	0.17 (0.329)
Chemo therapy	−0.02 (0.689)	0.17 (0.005)
Radiation therapy	0.03 (0.547)	−0.04 (0.534)
Feeling during sex-related communication with HCP		
Shame (I would be shameful)	0.07 (0.209)	0.11 (0.063)
Embarrassment (I would be embarrassed)	0.12 (0.042)	0.13 (0.033)
Unpleasure (Health professional would be unpleasant)	0.13 (0.027)	0.13 (0.032)

**Table 5 ijerph-18-03752-t005:** Influencing factors of sexual information needs in men and women.

Variables	Sexual Information Needs
	Adjusted Odds Ratio (95% Confidence Interval)
Men	
Job (ref, no)	-
Monthly income (ref, <3261 US dollars)	0.43 (0.25–0.73) **
Smoking (ref, no)	-
Shameful feeling (ref, low)	-
Embarrassment (ref, low)	0.87 (0.39–1.91)
Unpleasant to health professional (ref, low)	1.41 (0.57–3.45)
Women	
Age (ref, <48 years)	0.51 (0.31–0.86)*
Job (ref, no)	
Monthly income (ref, <4,340,000 Korea Won)	
Marital status (ref, married)	
Alcohol drinking (ref, no)	1.88 (1.12–3.16) *
Time lapse after diagnosis (ref, <17 month)	-
Chemotherapy (ref, no)	1.87 (1.12–3.12) *
Shame (ref, no)	-
Embarrassment (ref, no)	1.60 (0.78–3.32)
Unpleasure (ref, no)	1.65 (0.62–4.40)

* *p* < 0.005, ** *p* < 0.001.

## Data Availability

The data that support the findings of this study are available from the corresponding authors (Youngji Kim) upon reasonable request.

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
