# Peer review of "Gender Differences in Sexual Information Needs and Relating Factors in Cancer Patients: A Cross-Sectional Survey"

_ijerph, 2021, doi:10.3390/ijerph18073752_

Round 1

Reviewer 1 Report

ijerph-1129887 review

Gender differences in sexual information needs and relating factors in cancer patients: A cross-sectional survey

Recommendation: This paper focuses on a very important topic for clinical nurses- sexual health education for cancer patients. The paper has issues in clarity of writing, support for the study, some Methods areas, and the Discussion section, but I believe these can be addressed relatively easily. The topic is one that is not relevant to public environmental health nursing practice, education, and research; moreover, major issues with the paper are the lack of specific linkages to public health nursing theory, environmental health nursing, nursing education, or nursing practice particularly in the Discussion section.

TITLE & ABSTRACT The title describes the focus of the article and the study. The Abstract gives an adequate overview of the study.

Introduction There is no current cancer patients’ statistical status and literature review of gender differences affecting the sexual health of cancer patients. It might be useful to provide the differences revealed between male and female regarding sexual health needs, sexual health knowledge, and behaviors.

Line 42-43: Provide reference “Sexual health does not mean that there is no disease or impairment, but rather a physical, emotional, mental, and social wellbeing in relation to sexuality.”

METHODS Explain why the study had a large numbers of recruited participants for survey. Did the authors consider the vulnerability of cancer patients? If there was the reason for large scaled survey, the evidence of calculation of the number of subjects should be provided. Clarify the participants' including and excluding criteria. State how participants were recruited and by whom. Were incentives used for recruitment? How were the data collections delivered e.g., where, who served as instructor(s) and how were they trained?

Results Table 5 Provide more statistic informations on the logistic model-e.g. overall statistics, omnibus tests of model coefficients, Nagelkerke R square…

Discussion What are the nursing implications of relating factors in cancer patients in terms of gender differences? Provide a citation for the statement : line 236-238 “In addition, in the case of female cancer patients who consumed alcohol, the likelihood of expressing their desires or wishes relatively easily with increasing amount of alcohol they drank. related information is greater.”

CITATIONS AND REFERENCES   A few errors in format in citations and reference list entries- citation number. The references are not current (64% are dated 2014 or before), and most appear to be from peer-reviewed journals.

Author Response

Dear reviewer

Thank you very much for having considered our manuscript. We are very happy to have received a positive evaluation, and we would like to express our appreciation to  both reviewers for the thoughtful comments and helpful suggestions. Reviewer raised several concerns, which we have carefully considered and made every effort to address. We fundamentally agree with all the comments made by the Reviewers, and we have incorporated corresponding revisions into the revised manuscript. 
Red text indicates changes made according to Editor’s suggestions. We believe that our manuscript has been considerably improved as a result of these revisions, and hope that our revised manuscript is acceptable for publication in the International journal of environmental research and public health.
We would like to thank you once again for your consideration of our work and inviting us to submit the revised manuscript. We look forward to hearing from you.
Best regards,
Corresponding author

Reviewer 2 Report

This study aimed to identify the sexual information requirements and related factors according to gender to improve sexual health in cancer patients. The study of this question in oriental samples is relevant. Sample size is a strength of the study.

Introduction

Information is lacking about sexual information in cancer patients. There is no literature review on that question. There are no hypotheses.

Participants

It is striking that the participants were evaluated in the first semester of 2015: 6 years have passed! This issue should be addressed in Limitations.

Discussion

Gender differences in the need for sexual information require a more in-depth discussion. Does it have to do with sexual attitudes, the role of sexuality, sexual socialization?

Limitations

It is necessary to include the limitations of the study.

References

References are not listed

Author Response

(The authors gave the same response as above.)

Round 2

Reviewer 2 Report

Gender differences require reflection in the Discussion. There is literature on other diseases (e.g., myocardial infarction) that can help the authors to this reflection.

https://doi.org/10.1177/1054773812437241

https://doi.org/10.1016/j.amjcard.2012.01.355

https://doi.org/10.1161/CIRCULATIONAHA.114.012709

Author Response

Dear reviewer

We thank the reviewer for your careful reading of the manuscript and your constructive remarks. We have taken the comments on board to improve and clarify the manuscript. Please find below attached file to all comments.

Sincerely yours,
